# Photocatalytic Decomposition of Nitrobenzene in Aqueous Solution by Ag/Cu₂O Assisted with Persulfate under Visible Light Irradiation

**Wen-Shing Chen * and Jyun-Yang Chen**

Department of Chemical and Materials Engineering, National Yunlin University of Science & Technology, 123 University Road, Section 3, Douliou 640, Yunlin, Taiwan; m10915009@yuntech.edu.tw
* Correspondence: chenwen@yuntech.edu.tw; Tel.: +88-6553-42601 (ext. 4624); Fax: +88-6553-12071

**Abstract:** The mineralization of nitrobenzene was executed using an innovative method, wherein Ag/Cu₂O semiconductors stimulated by visible light irradiation were supported with persulfate anions. Batch-wise experiments were performed for the evaluation of effects of silver metal contents impregnated, persulfate concentrations and Ag/Cu₂O dosages on the nitrobenzene removal efficiency. The physicochemical properties of fresh and reacted Ag/Cu₂O were illustrated by X-ray diffraction analyses, FE-SEM images, EDS Mapping analyses, UV–Vis diffuse reflectance spectra, transient photocurrent analyses and X-ray photoelectron spectra, respectively. Because of intense scavenging effects caused by benzene, 1-propanol and methanol individually, the predominant oxidant was considered to be sulfate radicals, originated from persulfate anions via the photocatalysis of Ag/Cu₂O. As regards oxidation pathways, nitrobenzene was initially transformed into hydroxycyclohexadienyl radicals, followed with the production of 2-nitrophenol, 3-nitrophenol or 4-nitrophenol. Afterwards, phenol compounds descended from denitration of nitrophenols were converted into hydroquinone and *p*-benzoquinone.

**Keywords:** nitrobenzene; Ag/Cu₂O; persulfate; sulfate radical

## 1. Introduction

Nitrobenzene is commonly used for the manufacture of polyurethane by way of intermediates of aniline. It has been also applied in the following industries: plastics, pesticides, pharmaceuticals and explosives [1]. Due to high risks for mutagenicity and carcinogenicity suffered, effluent contaminated with nitrobenzene and related derivatives would cause strong damage to the aqueous circumstances [2,3]. Consequently, much effort has been focused on the development of economically and effectively treating manners for industrial wastewater.

Advanced oxidation processes have been extensively explored for the mineralization of nitrobenzene in wastewater due to its resistance to biodegradation resulted from the electron-withdrawing property of nitro groups [4]. Firstly, much research has been carried out on hydroxyl radical-based processes, such as Fenton's methods [5–7], Fenton-like manners [8–10], Fenton reagents with auxiliary ultrasound [11] and Fenton reagents coupled with fluidization flow patterns [12,13]. Secondly, as commercial titanium dioxide is under investigation, the significant enhancement of the nitrobenzene destruction efficiency took advantage of doping ferric oxides, which successfully prevent the combination of photon-induced electrons with holes [14]. In another aspect, ultraviolet absorbance band was obviously changed to the visible light range by virtue of impregnating ammonium and cerium nitrates simultaneously [15,16]. On the other hand, ozone supported with ultrasound was employed for nitrobenzene removal, wherein hydroxyl radicals were claimed to be predominant oxidizing agents [17,18]. The oxidation of nitrobenzene through the catalysis of ozone over aluminum silicate was performed to elucidate the influences of operating parameters [19,20]. Additionally, the electrochemical oxidation of nitrobenzene was

conducted by the modified $PbO_2$ electrode plates, which were incorporated with titanium dioxide nanotubes, resulting in ordered arrangement and surface areas extended [21,22].

To date, sulfate-radical-related processes have been also dedicated to decomposing nitrobenzene in wastewater. The thermal activation of persulfate was effective for the disposal of effluents contaminated with nitrobenzene, wherein reaction pathways consist of 2-nitrophenol, 4-nitrophenol, 2,6-dinitrophenol and 2,4-dinitrophenol [23]. Except ferrous ions for the catalytic transformation of persulfate to sulfate radicals, zinc metal and magnetized iron metal also exhibited fruitful performance on nitrobenzene removal [24–26]. With a view to promoting nitrobenzene abatement, experiments were carried out utilizing persulfate anions irradiated with ultraviolet light [27]. Even though persulfate activated with photoelectrons descended from semiconductors excited by visible light, this has been barely studied in regard to nitrobenzene oxidation. As expected, the band gap energy of semiconductors meets the luminous energy of the incident light, leading to the generation of photo-induced electron–hole pairs. Persulfate anions would be transformed into reactive sulfate radicals via the activation of photo-induced electrons [28].

In this research, an innovative technique for the effective removal of nitrobenzene in wastewater was developed. Considerable sulfate radicals could be produced by way of persulfate via photoelectrons originated from $Ag/Cu_2O$ irradiated with visible light, which are well known as semiconductors [29–32]. The influences of operating parameters on the nitrobenzene degradation behaviors were explored, such as persulfate concentrations and $Ag/Cu_2O$ dosages. Nitrobenzene decomposition pathways catalyzed by the $Ag/Cu_2O$ coupled with persulfate under visible light irradiation would be investigated in the meantime.

## 2. Experimental Methods

### 2.1. Testing of Photocatalytic Oxidation of Nitrobenzene by Ag/Cu₂O with Assistance of Persulfate

The experimental system containing the main equipment was referred to in our previous report [33]. The photocatalytic cell was a quartz cylinder fitted internally with a magnetic stirrer and cooling coils, in which the testing temperature was maintained through a thermostat (PIIN JIA Technol. Co. New Taipei City, Taiwan). Visible light irradiation was supplied from twelve lamps (8.6 W each) surrounding the cell with three chief peaks of 438 nm, 550 nm and 619 nm, respectively (Philips Corp. PL-S, Hanover, MD, US). Owing to consistence with practical concentrations of industrial wastewater, feedstock was prepared at 1.0 mM concentrations of nitrobenzene (≥99.5%, Riedel-de Haen, Seelze, Germany) [34], being well agitated with proper weights of sodium persulfate (≥99.5%, Fluka, Seelze, Germany) beforehand. The $Ag/Cu_2O$, manufactured from $Cu_2O$ powder (SHOWA, Tokyo, Japan) by incipient impregnation with 1–5 wt% of silver nitrate (≥99.5%, Fluka), respectively, and sequential 3 h calcination at 473 K by sieving with 400 mesh [35], was carefully loaded into the basket and fixed near the center of photocatalytic cell. For the duration of tests, samples were taken from the cell at constant time intervals, and sequentially quenched to the temperature of $273 \pm 0.5$ K to terminate nitrobenzene oxidation [36]. Aqueous samples were executed on total organic carbon analyses to evaluate residual organic compounds. The $Ag/Cu_2O$ separated from oxidation tests were examined by an X-ray photoelectron spectrometer. In this work, experiments were performed in a series of persulfate concentrations of 35.0 to 70.0 mM to elucidate the sulfate radical effect at the pH values of 5 to 6. Photocatalytic tests under diverse dosages of $Ag/Cu_2O$ (1.05 up to 1.50 g $L^{-1}$) were carried out for enhancement on nitrobenzene removal efficiency. All experiments were undertaken repeatedly for the affirmation of data reliability.

### 2.2. Total Organic Carbon (TOC) Analysis

Within the duration of photocatalytic testing by $Ag/Cu_2O$ assisted with persulfate, wastewater was periodically sampled and measured, utilizing a TOC analyzer (GE Corp. Sievers InnovOx, Boston, MA, USA). The hydrocarbons involved were completely oxidized into carbon dioxide and quantified through nondispersive infrared (NDIR) analyses,

wherein persulfate mineralization was carried out under supercritical water conditions. On the contrary, non-hydrocarbons were exhausted in the species of carbonic acid. In this work, the TOC concentrations reported were calibrated to the standard curve, fulfilled faithfully in the range (0–5.0 mM) utilizing standard chemicals of potassium hydrogen phthalate.

### 2.3. Physicochemical Properties of $Ag/Cu_2O$

The crystal compositions of fresh $Ag/Cu_2O$ semiconductors were examined using an X-ray diffractometer (Rigaku, MiniFlex II, Tokyo, Japan) integrated with high-intensity monochromated CuKα radiation at the wavelength of 0.15418 nm, operating with the current of 30 mA and 40 kV volts over the 2θ range from 10 to 80 degrees. For the inspection of surface morphology and silver content impregnated on $Ag/Cu_2O$ semiconductors, surface scanning was executed by a field-emission scanning electron microscope (JEOL, JSM-6500F) equipped with an energy dispersive X-ray spectroscope (JEOL, JED-2300). As far as light absorption band is concerned, the Ultraviolet–Visible diffuse reflectance spectra on $Ag/Cu_2O$ were measured using a UV–Vis spectrometer (PerkinElmer, Lambda-850, Waltham, MA, USA), of which wavelength was among the range of 250 to 800 nm with reference to $BaSO_4$. The photocurrent measurements of the samples were carried out using a potentiostat (Zensor, ECAS-100, Etterbeek, Belgium) under continuous visible light irradiation (Philips, PL-S), wherein a Pt wire served as an auxiliary electrode, coated with $Ag/Cu_2O$ and polymer electrolyte membrane, and was referenced to a saturated calomel electrode. Further, electronic states of fresh and reacted $Ag/Cu_2O$ semiconductors were monitored by means of XPS spectra from an X-ray photoelectron spectrometer (Kratos Analytical Ltd. Axis Ultra, Manchester, UK), in which monochromated AlKα irradiation was used as a light source and the binding energy of samples was calibrated to 284.8 eV for C 1s core level of adventitious carbon.

### 2.4. Gas Chromatography–Mass Spectrometry (GC-MS) Analysis

A proper amount of wastewater was withdrawn from the photocatalytic cell after nitrobenzene oxidation testing for 30 min. The microextraction fiber spread with carboxen/polydimethylsiloxane (Supelco, Bellefonte, PA, USA) was added into aqueous solution for the effective adsorption of degradation intermediates. Then, the fiber was directly packed into a micro-needle which was immediately injected into the orifice of the gas chromatograph–mass spectrometer (Hewlett Packard, MASS 59864B/5973, Palo Alto, CA, USA). The metal capillary column for ingredient separation was used at the dimension of 30 m × 0.25 mm (Ultra ALLOY UA-5), wherein helium gas served as the carrier gas. The degradation intermediates resolved were trustworthy based on mass spectra obtained in comparison with those of standards.

### 2.5. Scavenging Effects

In order to disclose the main oxidants on the mineralization of nitrobenzene, testing was conducted in the presence of diverse scavengers simultaneously, such as methanol, 1-propanol and benzene, respectively [37,38]. The nitrobenzene degradation percentage was determined on the basis of the peak area (262 nm) shown in a UV–Vis spectrophotometer (PerkinElmer, Lambda 850) [8]. In the course of pretesting, benzene was verified as the most sharp scavenger. The benzene scavenging effect may represent sulfate radical yields at different experimental conditions. To evaluate sulfate radical yields, the decrement of nitrobenzene degradation percentage was examined upon the addition of suitable amounts of benzene scavenger into wastewater.

## 3. Results and Discussion

### 3.1. Comparison of Photocatalytic Oxidation by $Ag/Cu_2O$ Alone and $Ag/Cu_2O$ Assisted with Persulfate Respectively

Figure 1 demonstrates the time flow patterns of TOC removal efficiency executed by photocatalytic oxidation over $Ag/Cu_2O$ alone and $Ag/Cu_2O$ aided with persulfate,

respectively. Apparently, the nitrobenzene removal rate caused by Ag/Cu$_2$O supported with persulfate process was much higher than those simply using persulfate anions, Cu$_2$O and Ag/Cu$_2$O alone. Noteworthily, Ag(1%)/Cu$_2$O integrated with persulfate displayed a synergistic performance in comparison with photocatalytic behaviors exhibited with the Ag(1%)/Cu$_2$O and persulfate individually. The observation could be ascribed to great enhancement on reactive sulfate radical yields. In fact, persulfate anions have been successfully transformed to sulfate radicals through the photocatalysis of Cu$_2$O [39]. It has been also recognized that Ag metal functions as an electron sink and strengthens charge separation, leading to the repression of the combination of photo-induced electrons and holes over Cu$_2$O [40,41]. As expected, a higher extent of Ag metal doped on the surface of Cu$_2$O gave rise to higher nitrobenzene removal efficiency (refer to Figure 2). The main reactions that occurred could be concluded as follows.

(1)     $Ag/Cu_2O + h\upsilon \rightarrow h^+{}_{vb} + e^-{}_{cb}$
(2)     $S_2O_8{}^{2-} + e^-{}_{cb} \rightarrow SO_4\bullet^- + SO_4{}^{2-}$
(3)     $SO_4{}^{2-} + h^+{}_{vb} \rightarrow SO_4\bullet^-$
(4)     $H_2O + h^+{}_{vb} \rightarrow HO\bullet + H^+$

where in $e^-{}_{cb}$ represents photo-induced electrons in the conduction band and $h^+{}_{vb}$ represents photo-induced holes in the valence band. Ag(5%)/Cu$_2$O was chosen as a candidate for next testing due to its better nitrobenzene degradation behavior.

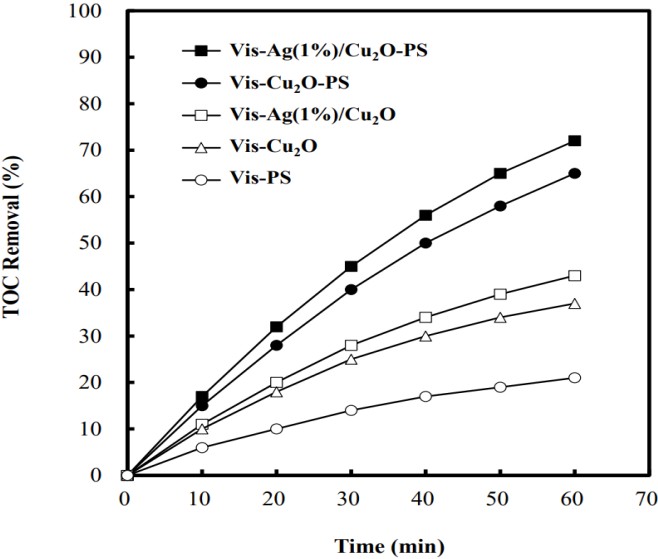

**Figure 1.** Time flow patterns of TOC removal efficiency by Cu$_2$O, persulfate, Ag(1 wt%)/Cu$_2$O, Cu$_2$O-persulfate and Ag(1 wt%)/Cu$_2$O-persulfate, respectively under the conditions of visible light power = 103.2 W, persulfate concentration = 50 mM, Cu$_2$O or Ag(1 wt%)/Cu$_2$O dosage = 1.20 g L$^{-1}$ and T = 318 K.

XPS measurements were carried out to elucidate surface electronic states of Ag(5%)/Cu$_2$O. Figure 3 illustrates the Cu2p XPS spectra of fresh Ag(5%)/Cu$_2$O and reacted Ag(5%)/Cu$_2$O. As far as fresh Ag(5%)/Cu$_2$O semiconductor is concerned, two peaks centered at 933.0 and 952.0 eV were clearly found, which were appointed to the binding energy of Cu$^+$2p $_{(3/2)}$ and Cu$^+$2p $_{(1/2)}$, respectively [42–44]. With regard to reacted Ag(5%)/Cu$_2$O, four peaks centered at 933.5, 941.0, 953.0 and 961.0 eV were present, which were separately assigned to the binding energy of Cu$^+$2p $_{(3/2)}$, Cu$^{2+}$2p $_{(3/2)}$, Cu$^+$2p $_{(1/2)}$ and Cu$^{2+}$2p $_{(1/2)}$ [45,46]. Obviously, Cu$^+$ cations on the surface of reacted Ag(5%)/Cu$_2$O shifted to higher oxidation states in comparison with the fresh one, in consideration of the migration of photo-induced electrons to persulfate anions [39,47]. The observations manifest the above hypothesis that persulfate anions could be converted into sulfate radicals upon activation by photo-induced electrons. Instead, sulfate anions may be also

transformed to sulfate radicals via photo-induced holes over Ag(5%)/Cu$_2$O [48]. It makes partial contributions for nitrobenzene oxidation.

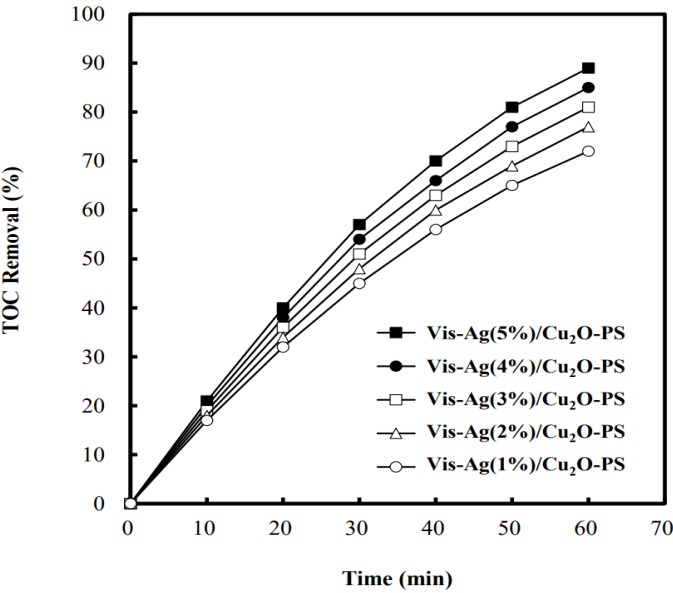

**Figure 2.** Effect of silver metal contents on the TOC removal efficiency under the conditions of visible light power = 103.2 W, persulfate concentration = 50 mM, Ag/Cu$_2$O dosage = 1.20 g L$^{-1}$ and T = 318 K.

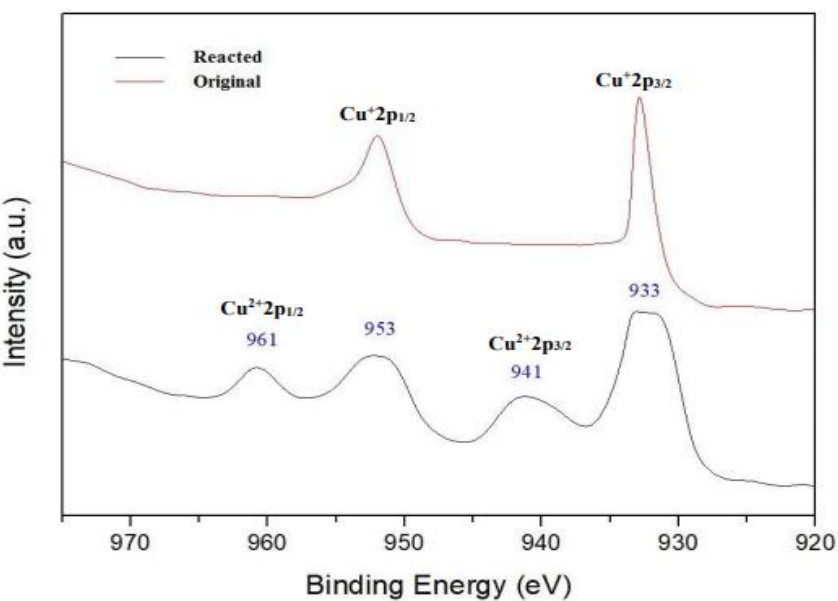

**Figure 3.** X-ray photoelectron spectra of Cu$^+$2p or Cu$^{2+}$2p core level for original Ag(5 wt%)/Cu$_2$O and reacted Ag(5 wt%)/Cu$_2$O semiconductors.

### 3.2. Physicochemical Properties of Ag/Cu$_2$O

The X-ray diffraction patterns of Ag/Cu$_2$O semiconductors are displayed in Figure 4. Major peaks in the spectra were matched with crystal planes of Cu$_2$O, wherein the characteristic peak at 2θ value of 36.5° was ascribed to the (111) plane [48,49]. Conversely, the diffraction peak at 2θ values of 38.1° was appointed to the (111) plane of Ag metal [50]. It is evident that slight weight of Ag metal was doped on the surface of Cu$_2$O. Figure 5 presents field-emission SEM images of Ag/Cu$_2$O semiconductors. Obviously, the majority of the Cu$_2$O surface was smooth. In contrast, a few irregular-shaped sediments were found over

Ag/Cu$_2$O and more clumps of particles were observed upon increasing Ag extent. It is apparent that Ag metal was well spread on the surface of Cu$_2$O. The Energy-Dispersive X-ray Mapping analysis on Ag/Cu$_2$O is illustrated in Figure 6. Ag metal was well dispersed, and its contents measured agreed with those impregnated theoretically (see Table 1).

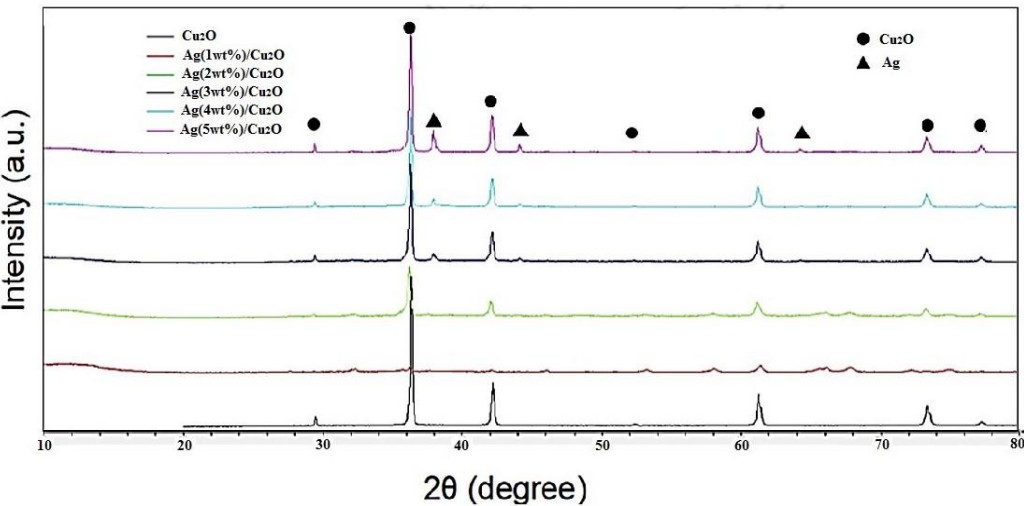

**Figure 4.** The XRD patterns of Cu$_2$O and Ag/Cu$_2$O semiconductors.

Figure 7 depicts the UV–Vis diffuse reflectance spectra for a series of Ag/Cu$_2$O semiconductors. The analogous spectra were observed between Ag/Cu$_2$O and Cu$_2$O at the absorbance wavelength between 400 and 530 nm, which fall into the visible light range. Particularly, the light absorbance intensity of Ag/Cu$_2$O was stronger than that of Cu$_2$O. It implies that Ag/Cu$_2$O semiconductors are more responsive to the visible light irradiation. This phenomenon could be mainly ascribed to Ag metal dopant, creating an electron sink and restraining a combination of photo-induced electrons with holes over Cu$_2$O [51,52]. Further, the Ag/Cu$_2$O band gap energy was resolved by means of a Tauc's equation $[(\alpha h\nu)^{1/n} = A(h\nu - E_g)]$, in which h$\nu$ stands for incident optical energy. The "n" parameter was set at the value of 1/2 according to the electronic transition state of Ag/Cu$_2$O semiconductors. The draft of $(\alpha h\nu)^2$ varied with incident optical energy; (h$\nu$) was drawn to obtain the band gap energy by intercepting tangent lines to the X-axis [53–55]. Consequently, the Cu$_2$O band gap energy was determined to be 2.17 eV, consistent with that reported by Muthukumaran et al. [56]. For a series of Ag/Cu$_2$O semiconductors, the band gap energy was estimated to be 2.06, 1.92, 1.75, 1.55 and 1.43 eV, respectively, upon increasing Ag metal doping (refer to Table 2). The superior photocatalytic performance presented by Ag(5%)/Cu$_2$O could be reasonably attributed to a significant yield of photo-induced electrons, caused by a lower band gap energy. In other words, the optical energy of visible light could stimulate Ag/Cu$_2$O semiconductors for the generation of electron–hole pairs. Persulfate anions would be effectively converted into reactive sulfate radicals via activation of photo-induced electrons. Likewise, photo-induced holes may also transform sulfate anions into sulfate radicals. Figure 8 illustrates transient photocurrence of Cu$_2$O and Ag(5%)/Cu$_2$O excited under visible light irradiation. The photocurrent intensity of the latter was clearly higher than that of the former. It means that Ag(5%)/Cu$_2$O possessed the higher yield of photo-induced electron [40,41]. The results support the issue of the inhibition of a combination of photo-induced electrons and holes over Cu$_2$O through the impregnation of Ag metal.

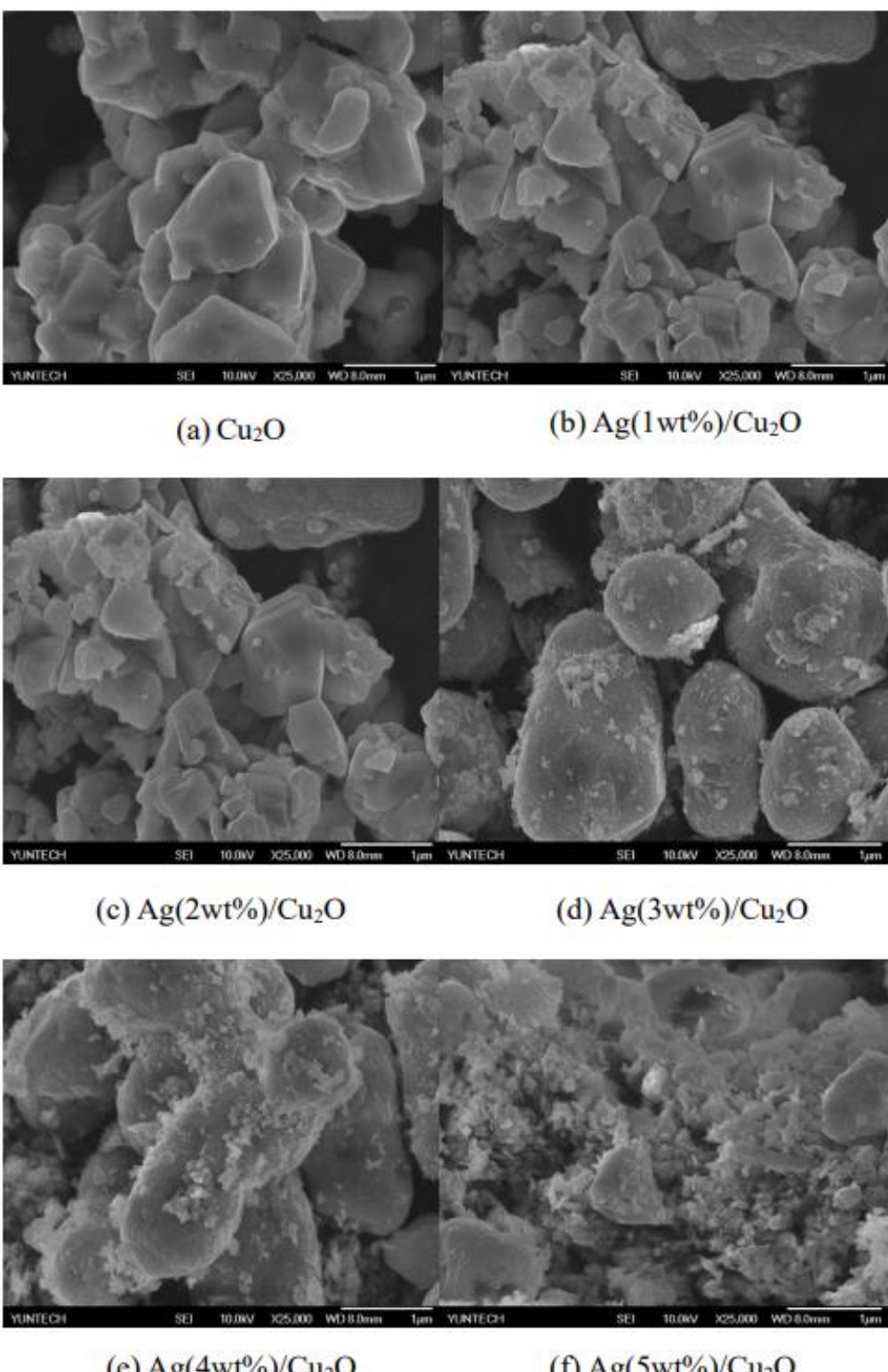

**Figure 5.** FE-SEM images of the (**a**) $Cu_2O$, (**b**) Ag(1 wt%)/$Cu_2O$, (**c**) Ag(2 wt%)/$Cu_2O$, (**d**) Ag(3 wt%)/$Cu_2O$, (**e**) Ag(4 wt%)/$Cu_2O$ and (**f**) Ag(5 wt%)/$Cu_2O$ semiconductors.

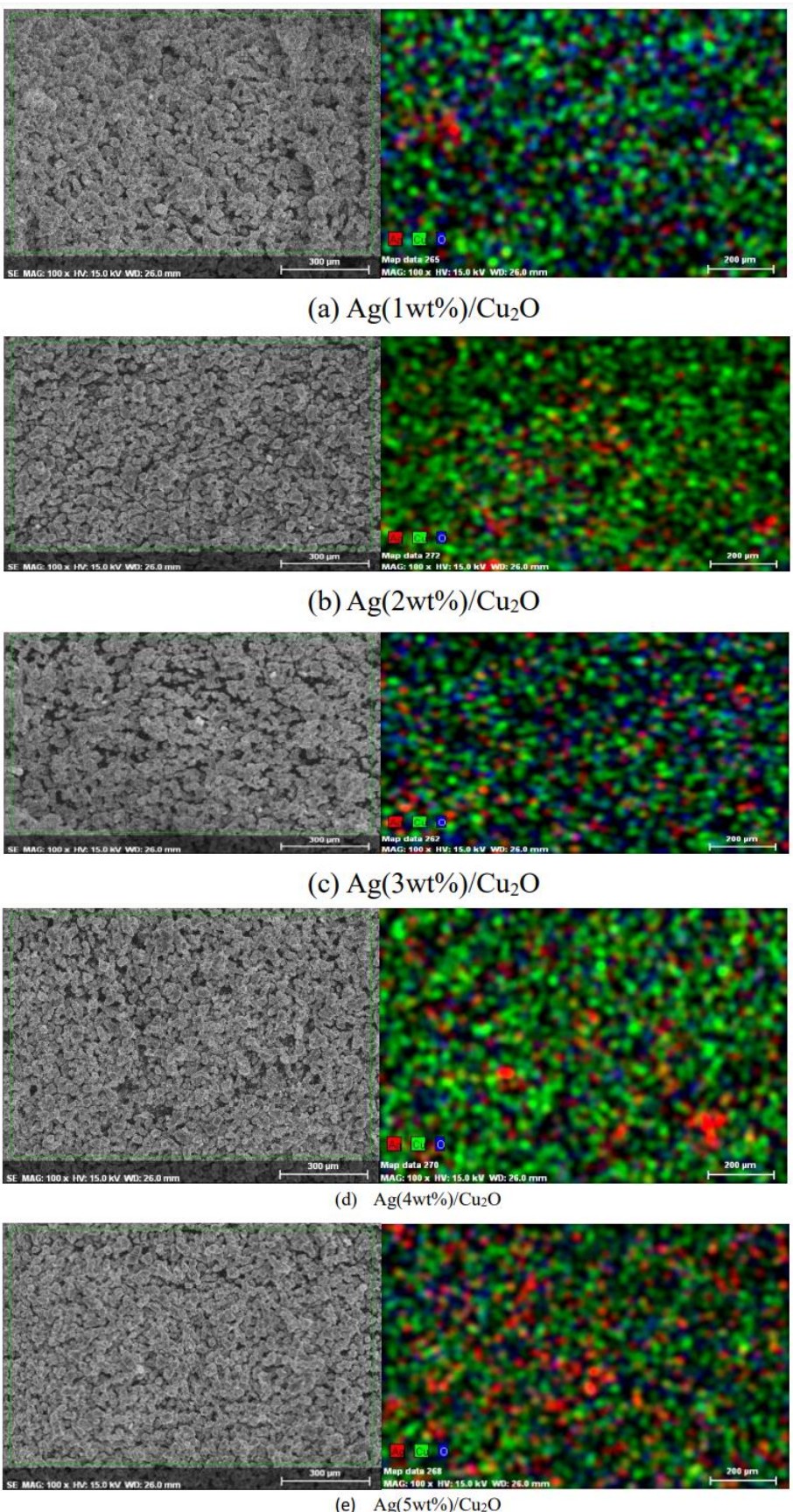

**Figure 6.** The EDS Mapping analyses on Ag/Cu$_2$O semiconductors: (**a**) Ag(1 wt%)/Cu$_2$O, (**b**) Ag(2 wt%)/Cu$_2$O, (**c**) Ag(3 wt%)/Cu$_2$O, (**d**) Ag(4 wt%)/Cu$_2$O and (**e**) Ag(5 wt%)/Cu$_2$O.

**Table 1.** The elemental compositions of semiconductors by EDS analyses.

| Semiconductor | Ag(wt%) | Cu(wt%) | O(wt%) |
|---|---|---|---|
| $Cu_2O$ | 0.00 | 85.81 | 14.19 |
| Ag(1 wt%)/$Cu_2O$ | 1.15 | 87.41 | 11.44 |
| Ag(2 wt%)/$Cu_2O$ | 2.05 | 87.56 | 10.39 |
| Ag(3 wt%)/$Cu_2O$ | 3.56 | 84.32 | 12.12 |
| Ag(4 wt%)/$Cu_2O$ | 4.99 | 82.39 | 12.62 |
| Ag(5 wt%)/$Cu_2O$ | 6.84 | 80.67 | 12.49 |

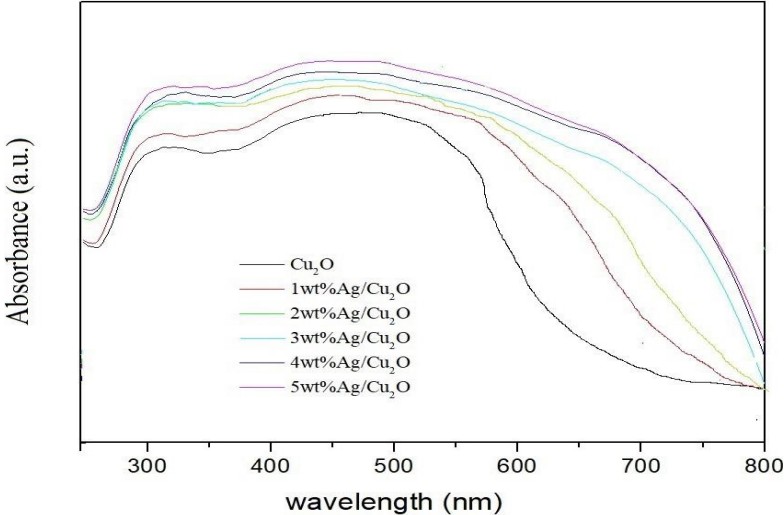

**Figure 7.** UV–Vis diffuse reflectance spectra of $Cu_2O$, Ag(1 wt%)/$Cu_2O$, Ag(2 wt%)/$Cu_2O$, Ag(3 wt%)/$Cu_2O$, Ag(4 wt%)/$Cu_2O$ and Ag(5 wt%)/$Cu_2O$ semiconductors.

**Table 2.** The band gap energy of semiconductors by UV–Vis DRS analyses.

| Semiconductor | Band Gap Energy (eV) |
|---|---|
| $Cu_2O$ | 2.17 |
| Ag(1 wt%)/$Cu_2O$ | 2.06 |
| Ag(2 wt%)/$Cu_2O$ | 1.92 |
| Ag(3 wt%)/$Cu_2O$ | 1.75 |
| Ag(4 wt%)/$Cu_2O$ | 1.55 |
| Ag(5 wt%)/$Cu_2O$ | 1.43 |

*3.3. Effect of Scavenger Dosages on Photocatalytic Oxidation by Ag/$Cu_2O$ Assisted with Persulfate*

Equivalent concentrations of benzene, 1-propanol and methanol were individually blended with nitrobenzene in wastewater to disclose reactive radicals under photocatalysis by Ag/$Cu_2O$ with assistance of persulfate. As demonstrated in Figure 9, nitrobenzene removal efficiency was sharply faded upon the addition of benzene, due to a high reaction rate constant for benzene and sulfate radicals ($3 \times 10^9$ M$^{-1}$ s$^{-1}$) [37]. Alternatively, 1-propanol and methanol slightly suppressed the nitrobenzene removal rate, on account of moderate rate constants for 1-propanol and methanol, with sulfate radicals being $6.0 \times 10^7$ M$^{-1}$ s$^{-1}$ and $3.2 \times 10^6$ M$^{-1}$ s$^{-1}$, respectively [57]. It deserves noting that the extent of nitrobenzene removal percentages faded corresponds to the reactive activity for various scavengers and sulfate radicals. It was revealed that sulfate radicals were principal oxidants for nitrobenzene degradation in wastewater.

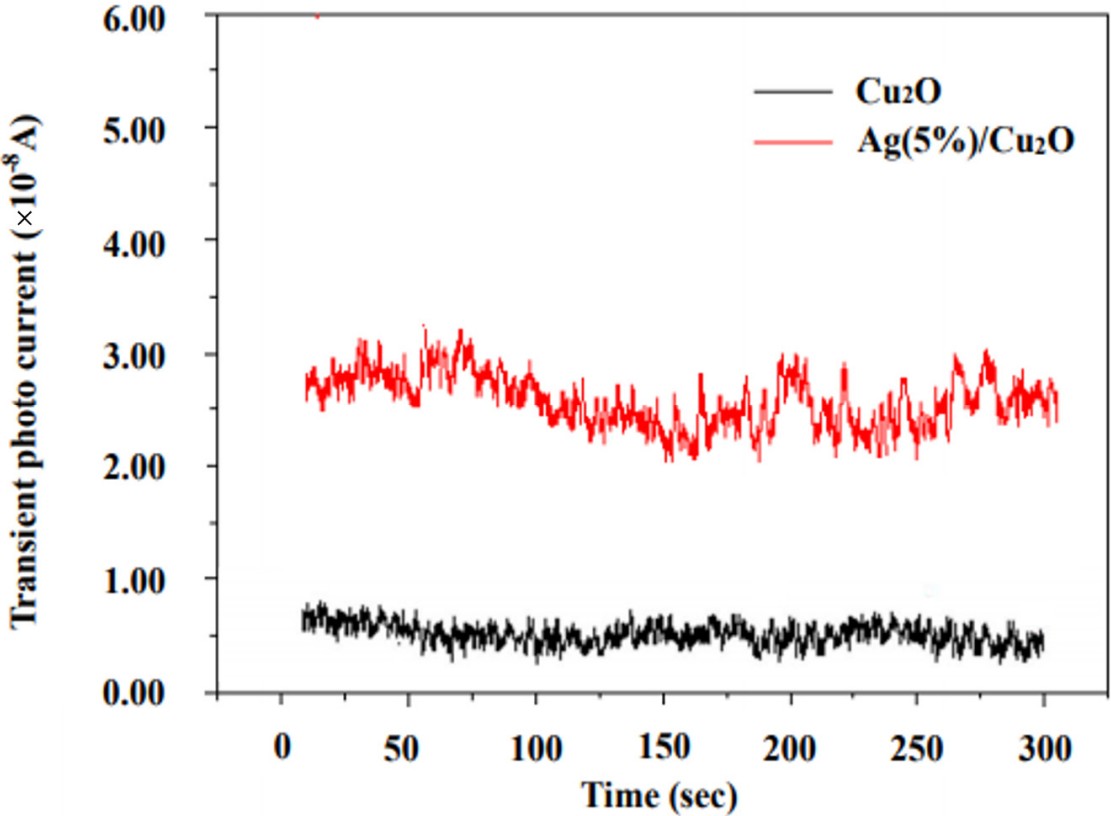

**Figure 8.** The transient photocurrent analyses of $Cu_2O$ and $Ag(5\%)/Cu_2O$ under visible light irradiation.

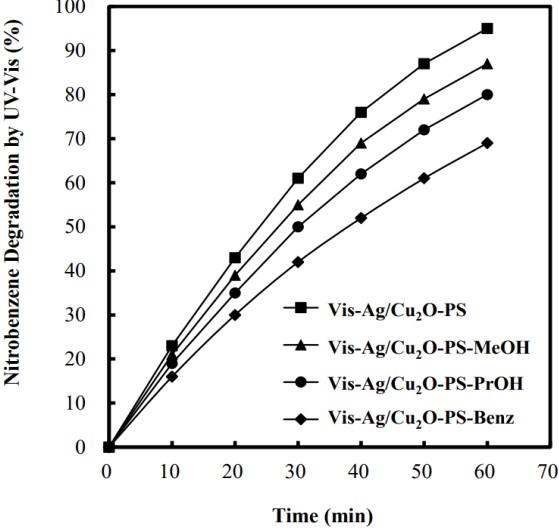

**Figure 9.** Effect of coexistence of benzene, 1-propanol and methanol, respectively, on the nitrobenzene degradation efficiency.

### 3.4. Effect of Persulfate Concentrations on Photocatalytic Oxidation by $Ag/Cu_2O$ Assisted with Persulfate

The optimal persulfate concentration for nitrobenzene elimination should be determined in consideration of commercialization. As presented in Figure 10a, the time flow patterns of TOC removal efficiency were dependent on persulfate concentrations. Undoubtedly, increasing persulfate concentrations enhanced nitrobenzene removal rates, wherein high sulfate radical yields could be sensibly expected. Nonetheless, the nitrobenzene removal efficiency faded under an excess persulfate concentration (70 mM). This phenomenon

may be interpreted with probable side reactions for the overdosage of persulfate anions and sulfate radicals [38,58]. Further, the photocatalytic oxidation of nitrobenzene was performed in the existence of benzene scavengers to discriminate sulfate radicals yields under various persulfate concentrations (refer to Figure 10b). Definitely, the scavenging effect dramatically displays an analogy between sulfate radical yields and TOC removal efficiency. Accordingly, sulfate radicals were likely to be responsible for nitrobenzene oxidation.

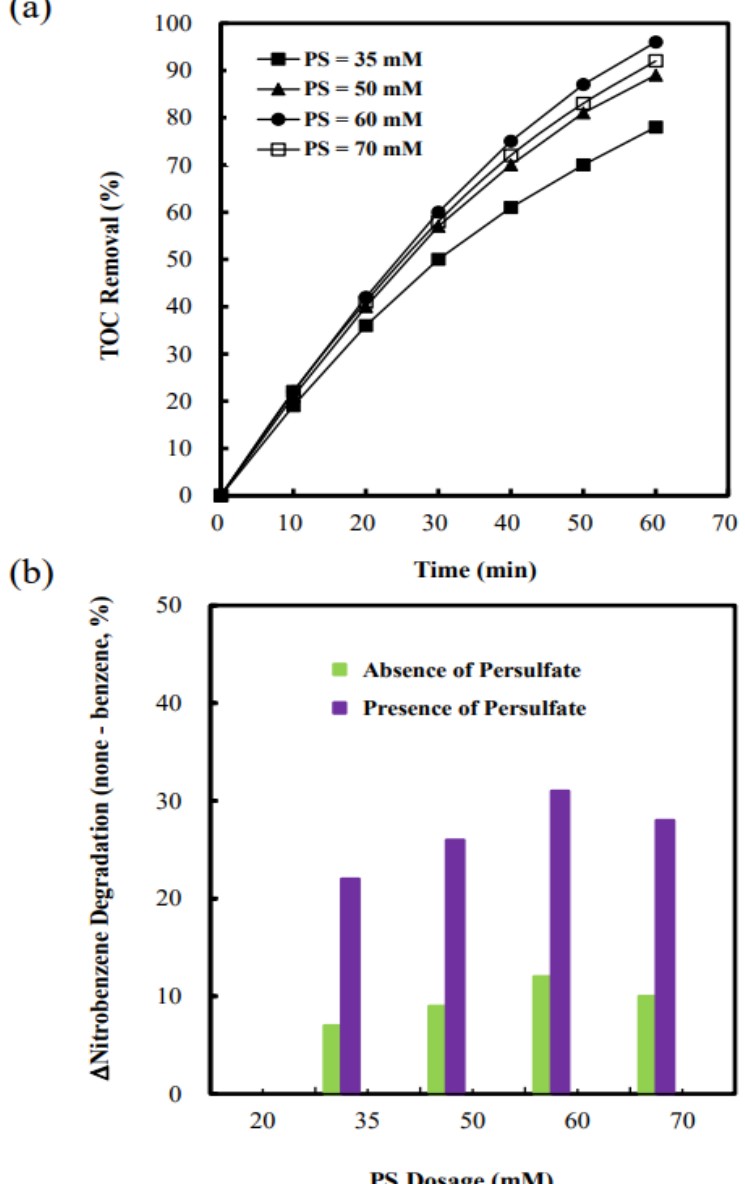

**Figure 10.** (**a**) Effect of persulfate concentrations on the TOC removal efficiency under the conditions of visible light power = 103.2 W, Ag(5 wt%)/Cu$_2$O dosage = 1.20 g L$^{-1}$ and T = 318 K. (**b**) The difference of nitrobenzene degradation efficiency between the absence of benzene and presence of benzene monitored by UV–Vis and served as scavenging effect under the conditions of visible light power = 103.2 W, Ag(5 wt%)/Cu$_2$O dosage = 1.20 g L$^{-1}$ and T = 318 K.

### 3.5. Effect of Ag/Cu$_2$O Dosage on Photocatalytic Oxidation by Ag/Cu$_2$O Assisted with Persulfate

An optimal dosage of Ag/Cu$_2$O semiconductor needs to be essentially established from the process design viewpoint. Figure 11a illustrates the time flow patterns of TOC removal efficiency as functions of Ag/Cu$_2$O dosages. Evidently, the nitrobenzene degradation rate raised with an increment of Ag/Cu$_2$O dosages, whereas it reduced under

overdosages of $Ag/Cu_2O$ ($\geq 1.50$ g $L^{-1}$). The improvement in the nitrobenzene removal efficiency could be attributed to high sulfate radical yields, caused by the massive activation of persulfate anions via the photocatalysis of $Ag/Cu_2O$. Conversely, lesser semiconductors received optical energy because of scatter of visible light irradiation, resulting from excessive dosages of $Ag/Cu_2O$ powder [59]. Nitrobenzene decomposition efficiency likewise displayed a similar trend as benzene scavenging behaviors (refer to Figure 11b). The outcomes convince us that sulfate radicals were chief oxidants toward nitrobenzene destruction. Especially, the optimal conditions for complete mineralization of nitrobenzene were determined as follows: visible light power = 103.2 W, persulfate concentration = 60 mM, $Ag/Cu_2O$ dosage = 1.35 g $L^{-1}$ and T = 318 K. In this work, the photocatalytic stability of $Ag/Cu_2O$ was proved via repetitions of five tests (shown in Figure 12). Evidently, nitrobenzene removal efficiency reached almost 98% during the overall experiment. That convinces us of the feasibility for the potential application of $Ag/Cu_2O$ to industrial wastewater treatment.

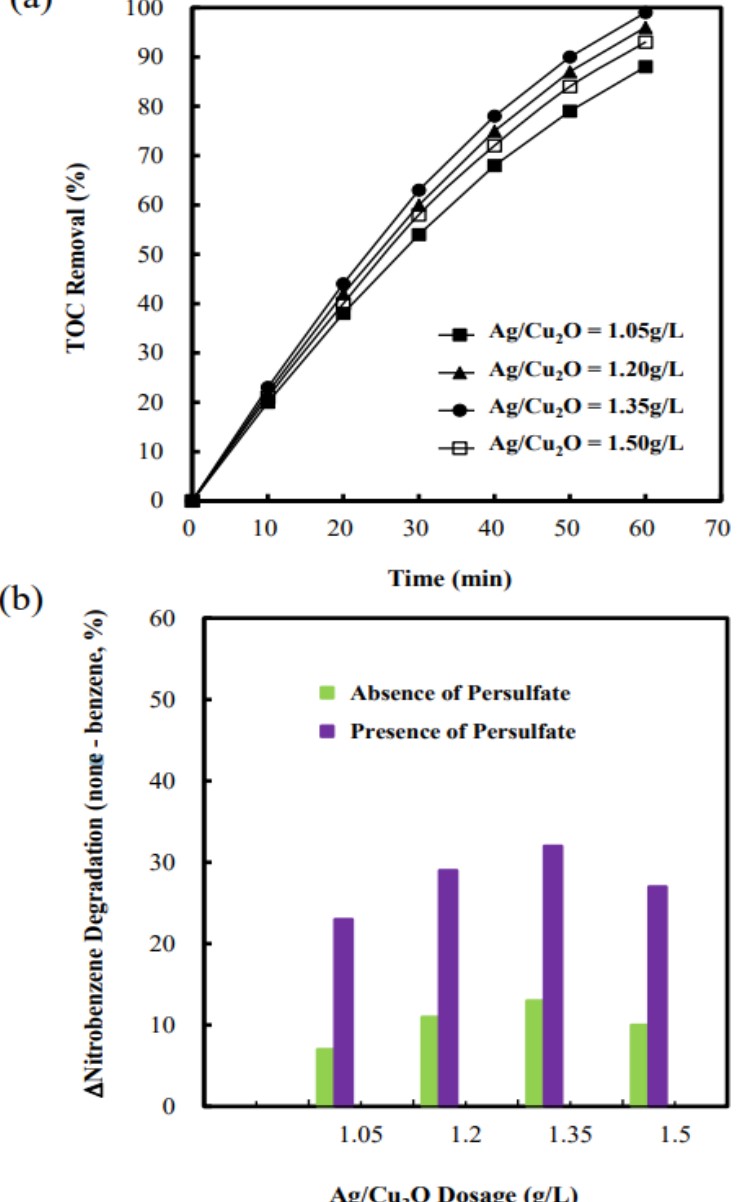

**Figure 11.** (**a**) Effect of Ag(5 wt%)/$Cu_2O$ dosages on the TOC removal efficiency under the conditions of visible light power = 103.2 W, persulfate concentration = 60 mM and T = 318 K. (**b**) The difference

of nitrobenzene degradation efficiency between the absence of benzene and presence of benzene monitored by UV–Vis and served as scavenging effect under the conditions of visible light power = 103.2 W, persulfate concentration = 60 mM and T = 318 K.

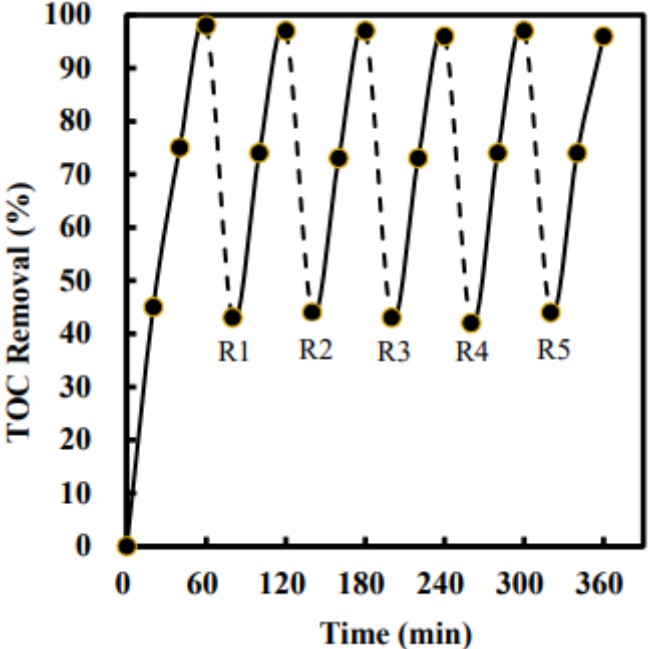

**Figure 12.** The photocatalytic stability of Ag(5 wt%)/$Cu_2O$ examined by means of repetitions of five tests.

### 3.6. Reaction Pathways of Photocatalytic Oxidation of Nitrobenzene by Ag/$Cu_2O$ Assisted with Persulfate

All reaction intermediates extracted from photocatalytic oxidation of nitrobenzene using Ag/$Cu_2O$ with assistance of persulfate were examined by a GC-MS spectrometer. Table 3 summarizes the ingredients procured, including nitrobenzene used for raw materials, phenol, 2-nitrophenol, 3-nitrophenol, 4-nitrophenol, hydroquinone and *p*-benzoquinone. In consideration of the source of 2-nitrophenol, 3-nitrophenol and 4-nitrophenol, it was believed that nitrobenzene underwent $O_2$ addition, followed with $HO_2$• elimination for the generation of hydroxycyclohexadienyl radicals and sequential hydroxylated compounds [57,60]. Phenol was clearly monitored as an intermediate because of the probable occurrence of nitrophenol denitration [61]. Afterward, phenol ordinarily executed oxidation reaction to hydroquinone, accompanied with successive hydrogen abstraction to *p*-benzoquinone. Ultimately, nitrobenzene would be mineralized into nitrate ions (analyzed by UV–Vis 313 nm), water and carbon dioxide. Based on most degradation intermediates cautiously identified, the hypothesized pathways for photocatalytic oxidation of nitrobenzene by Ag/$Cu_2O$ aided with persulfate is demonstrated in Figure 13.

**Table 3.** Compositions of nitrobenzene and reaction intermediates identified by GC-MS.

| Component | m/z (Relative Abundance, %) |
|---|---|
| **Feedstock** | |
| Nitrobenzene | 50 (15.7), 51 (37.7), 65 (13.6), 74 (9.0), 77 (100), 78 (7.5), 93 (16.9), 123(70.2) |
| **Reaction intermediate** | |
| Phenol | 38 (5.4), 39 (12.5), 40 (6.9), 55(6.4), 63 (6.5), 65 (21.0), 66 (27.4), 94 (100), 95 (7.7) |
| 2-Nitrophenol | 39 (15.7), 53 (9.8), 63 (20.2), 64 (13.9), 65 (25.5), 81 (19.6), 109 (18.2), 139 (100) |
| 3-Nitrophenol | 39 (35.9), 53 (10.7), 63 (14.7), 64 (7.9), 65 (63.7), 81 (15.8), 93 (51.4), 139 (100) |
| 4-Nitrophenol | 39 (44.2), 53 (23.3), 63 (28.1), 65 (79.9), 81 (33.0), 93 (26.9), 109 (67.1), 139 (100) |
| Hydroquinone | 39 (6.9), 53 (14.4), 54 (12.9), 55 (10.5), 81 (25.3), 82 (12.3), 110 (100), 143 (9.6) |
| *p*-Benzoquinone | 26 (18.1), 52 (17.9), 53 (17.1), 54 (63.3), 80 (28.2), 82 (36.3), 108 (100), 110 (12.1) |

**Figure 13.** Overall reaction pathways of nitrobenzene in wastewater by photocatalysis of Ag/Cu$_2$O with assistance of persulfate.

## 4. Conclusions

In light of the above discussions, nitrobenzene contaminants were principally mineralized via reactive sulfate radicals, induced from persulfate anions activated effectively by the photocatalysis of Ag/Cu$_2$O semiconductors. It was intensely supported by benzene scavenger, wherein sulfate radical yields display an analogy with nitrobenzene removal efficiency. As far as GC-MS analyses are concerned, the overall reaction pathways on nitrobenzene oxidation could be proposed as follows. Firstly, nitrobenzene was transformed into hydroxycyclohexadienyl radicals, followed with oxidation step into 2-nitrophenol, 3-nitrophenol and 4-nitrophenol separately. Sequentially, nitrophenol-related compounds executed the denitration procedure to phenol, which was oxidized further for the synthesis of hydroquinone and *p*-benzoquinone. As expected, nitrobenzene would be nearly mineralized into carbon dioxide, nitrate ions and water. The striking results persuade us that the photocatalysis of Ag/Cu$_2$O coupled with persulfate is an effective and synergistic manner for disposal of industrial effluents.

**Author Contributions:** Conceptualization, W.-S.C. and J.-Y.C.; methodology, J.-Y.C.; software, J.-Y.C.; validation, W.-S.C., J.-Y.C.; formal analysis, J.-Y.C.; investigation, W.-S.C.; resources, W.-S.C.; data curation, J.-Y.C.; writing—original draft preparation, W.-S.C.; writing—review and editing, W.-S.C.; visualization, J.-Y.C.; supervision, W.-S.C.; project administration, W.-S.C.; funding acquisition, W.-S.C. All authors have read and agreed to the published version of the manuscript.

**Funding:** This research received no external funding.

**Institutional Review Board Statement:** Not applicable.

**Informed Consent Statement:** Not applicable.

**Data Availability Statement:** The study did not contain any data reported.

**Conflicts of Interest:** The authors declare no conflict of interest.

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
