# Peer review of "Photocatalytic Decomposition of Nitrobenzene in Aqueous Solution by Ag/Cu2O Assisted with Persulfate under Visible Light Irradiation"

_2673-7256, doi:10.3390/photochem1020013_

Round 1
Reviewer 1 Report
This paper reports the photodecomposition of nitrobenzene by Cu2O/Ag and persulfate ion. The experiments appear to have been very carefully performed, and the author’s explanation seems reasonable.
Thus, this might be acceptable for Photochem. Nevertheless, the author should make a few modifications/additions, as indicated at the end of this text.
- Did you examine the effect of oxygen or photoreaction under Ar? According to this, what is oxidant in the oxidation process of phenol under these conditions in Figure 12?
- In Figure 12, please revise radical anion of sulfate in the first step of nitrobenzene.
Author Response
Reviewer 1:
This paper reports the photodecomposition of nitrobenzene by Cu2O/Ag and persulfate ion. The experiments appear to have been very carefully performed, and the author’s explanation seems reasonable.
Thus, this might be acceptable for Photochem. Nevertheless, the author should make a few modifications/additions, as indicated at the end of this text.
Ans: Thank you very much for your kind instructions. According to the questionnaire listed, the revision and/or response are as follows:
- Did you examine the effect of oxygen or photoreaction under Ar? According to this, what is oxidant in the oxidation process of phenol under these conditions in Figure 12?
Ans: We did not examine the effect of oxygen dissolved on the nitrobenzene removal efficiency. In fact, the effect of reaction temperature on the nitrobenzene removal efficiency has been carried out. Higher temperature is beneficial for nitrobenzene elimination, whereas oxygen solubility in wastewater decreased at high temperature. Thus, sulfate radicals may be principal oxidants for nitrobenzene removal. Based on the results of scavenging experiments, the nitrobenzene removal efficiency faded corresponds to reactive activity of various scavengers with sulfate radicals. That provides evidence on main oxidants to be sulfate radicals.
In Figure 12, sulfate radicals are also chief oxidants for phenol oxidation. The oxidation mechanism is analogous to that on first step of nitrobenzene oxidation, wherein the hydroxylated compound would be formed.
- In Figure 12, please revise radical anion of sulfate in the first step of nitrobenzene.
Ans: In Figure 12, the chemical formula of sulfate radical anion has been revised as reviewer kindly suggested.
Reviewer 2 Report
In this manuscript, Chen et al, discussed Ag/Cu2O pf photocatalytic nitrobenzene degradation. The experiment design and discussion are well structured. After considering some revision, I suggest to publish this manuscript.
1) Please put what pH used in this experimental.
2) Please show what elements each of blue, green, and red in Fig. 6 corresponds to. Also, the scale is hard to see.
3) Please describe in detail the experimental conditions (did you use some substrate?, excitation light wavelength, how the current was measured?) in Fig. 8. Does ''Transient'' irradiate visible light 0 seconds before, then stop visible light and measure the photocurrent generated by some experimental condition?
4) If possible, stability of photocatayst should provide.
Author Response
Reviewer 2:
In this manuscript, Chen et al, discussed Ag/Cu2O for photocatalytic nitrobenzene degradation. The experiment design and discussion are well structured. After considering some revision, I suggest to publish this manuscript.
Ans: Thank you very much for your kind instructions. According to the questionnaire listed, the revision and/or response are as follows:
- Please put what pH used in this experiment.
Ans: The pH value operated has been indicated in the experimental method section.
- Please show what elements each of blue, green, and red in Fig. 6 corresponds to. Also, the scale is hard to see.
Ans: Figure 6 has been revised as reviewer kindly suggested.
- Please describe in detail the experimental conditions (did you use some substrate?, excitation light wavelength, how the current was measured?) in Fig. 8. Does ''Transient'' irradiate visible light 0 seconds before, then stop visible light and measure the photocurrent generated by some experimental condition?
Ans: The experimental procedures for photocurrent measurement have been described in detail at the experimental method section. The visible light irradiation emitted continuously during the experiment, wherein a steady photocurrent was detected continuously.
- If possible, stability of photocatayst should provide.
Ans: The photocatalytic stability of Ag/Cu2O has been examined for repetitions of five tests. Figure 12 illustrates the stability of Ag/Cu2O during the overall experiment.
Round 2
Reviewer 2 Report
Accept.